# Fuzzy Mathematics-Based Outer-Loop Control Method for Converter-Connected Distributed Generation and Storage Devices in Micro-Grids

Lorena Castro [1,†], Maximiliano Bueno-López [2,*,†] and Juan Mora-Flórez [1,†]

1   Department of Electric Power Engineering, Universidad Tecnológica de Pereira, Pereira 660003, Colombia; lcastro@utp.edu.co (L.C.); jjmora@utp.edu.co (J.M.-F.)
2   Department of Electronics, Instrumentation, and Control, Universidad del Cauca, Popayán 190001, Colombia
*   Correspondence: mbuenol@unicauca.edu.co
†   These authors contributed equally to this work.

**Abstract:** The modern changes in electric systems present new issues for control strategies. When power converters and distributed energy resources are included in the micro-grid, its model is more complex than the simplified representations used, sometimes losing essential data. This paper proposes a unified fuzzy mathematics-based control method applied to the outer loop of a voltage source converter (VSC) in both grid-connected and islanded modes to avoid using simplified models in complex micro-grids and handle the uncertain and non-stationary behaviour of nonlinear systems. The proposed control method is straightforwardly designed without simplifying the controlled system. This paper explains the design of a fuzzy mathematics-based control method applied to the outer-loop of a VSC, a crucial device for integrating renewable sources and storage devices in a micro-grid. Simulation results validated the novel control strategy, demonstrating its capabilities for real field applications.

**Keywords:** micro-grid; voltage source converter (VSC); fuzzy-logic control (FLC); outer-loop control

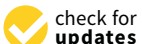



## 1. Introduction

### 1.1. Motivation

Micro-grids improve the reliability of electrical systems, supplying power to connected or islanded networks incorporating distributed generation, and these contribute to reducing pollution and integrating renewable energy resources [1]. However, there are many technical challenges regarding the integration of micro-grids into conventional distribution systems [2], including power flow control between the grid and the power converter [3], power management [4], voltage and frequency stability in grid-islanded mode [5–7], fault detection, and power quality, among others [8,9]. Most of the papers in the leading scientific databases do not address the transitions between the micro-grid operating modes; only one operation mode (connected or islanded from the main grid) is normally analysed, considering specific priorities [1,10]. Power converter control is a complex task due to the inherent nonlinearity. There is intensive research working toward developing newer control strategies to ensure output voltage stability under variations in load size, input voltage, and system parameters [11]. This paper aims to present the possibility of using an FLC to control different micro-grid variables, such as voltage, frequency, active power, and reactive power [12]. This paper contributes to demonstrating the capabilities of artificial intelligence in the control modules of converters in micro-grids.

### 1.2. State-of-the-Art

Fuzzy mathematics-based control is an attractive method because its structure, consisting of fuzzy sets and "if-then" rules, resembles how humans intuitively approach a

control problem. This makes it easier for a designer to incorporate heuristic knowledge of a system into the controller. Among the used control techniques, fuzzy mathematics-based approaches have been widely used in distribution systems, mainly to deal with the uncertainties originated through the inclusion of renewable and distributed energy sources [13,14]. As an alternative, fuzzy-based control (FLC) was proposed to avoid detailed modelling requirements but ensure high performance. One of the significant applications of fuzzy control methods in micro-grids is focused on the use of active disturbance rejection control (ADRC) as a representative data-driven (or model-free) control algorithm in order to exploit the advantages of data-driven control and fuzzy control [15,16].

Several applications are presented next: in [17,18], a general-purpose fuzzy controller for dc–dc converters was analysed. Additionally, [19] analysed a fuzzy-based control method for the maximum-power point tracking operation of a photo-voltaic system. A review presented in [20] described FLC applied in converters to minimise the photo-voltaic source output fluctuations, which produced undesirable effects in terms of harmonics output, power factor, switching schemes, and power losses. Likewise, in [21] the authors discussed the control of a three-phase, series, hybrid active filter connected to a photo-voltaic system aimed to minimise the sags, swells, and harmonics caused due to nonlinear power electronic loads. Similarly, the authors of [22] presented a rule-based nonlinear control method to deal with power converters. Finally, FLC was applied in [23] to design the outer loop of a converter only operating in grid-connected mode, using a single input to describe the system behaviour.

The effective integration of distributed energy resources into micro-grids is usually performed using power converters, where control strategies are required. The classical proportional-integral approach (PI) has been frequently used [3,24]. Additionally, in the case of grid-connected micro-grids, the Park's $dq$-axes transformation is considered in the control [25]. However, most previous approaches require simplified micro-grid models and significant effort to tune each converter's controller. Adequate performance is not always guaranteed, especially in cases of multiple converter-connected distributed energy resources [26,27].

### 1.3. Contributions

Fuzzy mathematics-based approaches have been widely used in electric systems by different types of experts. The contribution presented in this paper is a novel strategy focused on the control of grid-forming and grid-following converters, aimed at integrating distributed energy resources into micro-grids using fuzzy logic. Specifically, this paper presents a fuzzy-based outer-loop control method, as it is considered the most challenging yet settable part of the internal control (zero level control) of a VSC. Additionally, a comparison of the proposed fuzzy-based control for the outer-loop of a VSC was performed by considering a classical PI strategy based on Park's $dq$-axes transformation.

In summary, the main contributions of the proposed approach are:

(a) A strategy for efficiently developing an adjustable controller based on expert knowledge, represented by rules and memberships functions.

(b) A strategy that considers the complete system model, represented by rule-base and inference systems.

(c) A broad operating range controller to deal with perturbations in nonlinear systems.

(d) A control method that overcomes uncertainties associated with system parameter estimation.

(e) A faster control method than conventional approaches.

(f) A compressible strategy for quickly developing a well-performing controller for grid-forming and grid-following converters in micro-grid applications.

The fuzzy mathematics-based outer-loop controller has been proposed to replace the PI controller. As its main advantages, the former avoids using the mathematical converter model, adequately works with inaccurate inputs, and is robust. However, the

designed controller requires substantial computational effort due to complex and heuristic decision-making processes.

Finally, the fuzzy mathematics-based controller is robust as it reduces the complexity of the traditional control design and handles nonlinear systems, avoiding extensive mathematical modelling requirements.

### 1.4. Paper Organization

This paper is organised as follows: Section 2 presents the basic theoretical aspects of VSC and FLC; Section 3 presents the proposed methodology to design the outer-loop FLC; in Section 4, the testing scenarios, results, and analysis are addressed. Finally, the most important conclusions are highlighted in Section 5.

## 2. Fundamental Background

### 2.1. VSC Control

A micro-grid is a small-scale grid, which includes distributed generation, storage devices, and loads. Power converters are a fundamental part of micro-grids, and depending on the operating mode, these have different control objectives. In grid-connected mode, active ($P$) and reactive ($Q$) power management is controlled, while during grid-islanded mode, the interest is focused on control voltage ($v$) and frequency ($f$) [28,29].

VSCs are normally used to integrate distributed energy resources into the network. The zero-level controller is aimed to modify the pulses of IGBTs, and in this way, to obtain a desired value of the controlled variables [30]. In grid-connected mode, the controlled variables are $P$ and $Q$ for distributed resources, while in grid-islanded mode, it is oriented to control $v$ and $f$. Typically, the zero-level control method includes a phase-locked loop, a current control (inner-loop), pulse width modulation (PWM), and a power control or a voltage and frequency control (outer-loop) [3,9].

Outer-loop defines the reference currents in the direct and quadrature axis for the inner-loop. Therefore, the outer-loop is in charge of interest variable management according to the micro-grid operating mode [3,31].

Figure 1 shows the basic structure of a VSC in grid-connected or grid-islanded modes, according to the position of S1, S2, and S3.

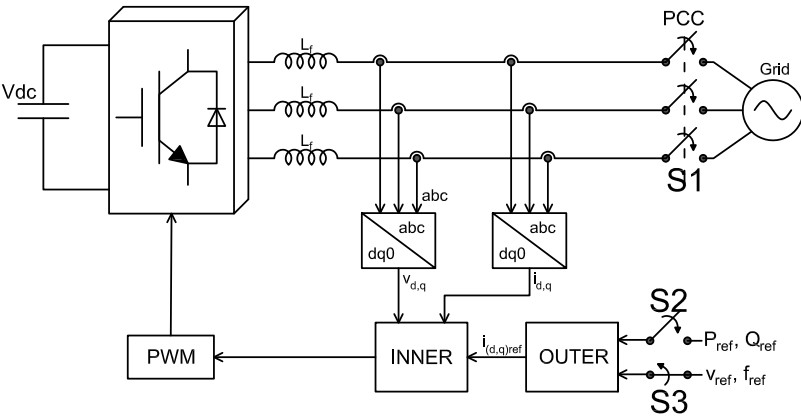

**Figure 1.** Scheme of VSC control considering microgrid operating modes.

### 2.2. General Structure of the Fuzzy-Based Control Method

As presented in Section 1, fuzzy mathematics has been widely applied in control systems because it provides a formal and straightforward methodology to represent, manipulate and include human expert heuristic knowledge to solve several engineering problems [32,33].

The FLC considered in this paper has the structure shown in Figure 2; it contains two inputs ($x$) and one output $y$, in a MISO structure (multiple input single output). Initially, FLC requires a fuzzification stage which is defined as a mapping from the acquired variables (input space) to a fuzzy representation (fuzzy sets); these sets are defined in a range between the maximum and minimum values allowed by each variable (universe of discourse). They require an inference mechanism to perform a nonlinear mapping from the input to the output spaces. This process involves all membership functions, fuzzy logical operators, and rules. Finally, a defuzzification process performs fuzzy control actions and delivers them as actions to be applied to the output system (actuators) [34,35].

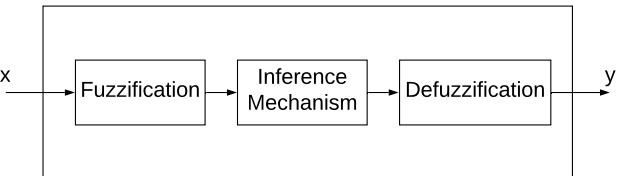

**Figure 2.** Structure of the proposed fuzzy logic based control.

## 3. Proposed Methodology

The methodology proposed to design the FLC for a VSC outer-loop controller is presented in Figure 3. This proposal is divided into three stages, which starts by analysing the system operation modes and ends by defining the control strategy. FLC allows considering a complete system model without using linear representations to calculate the control constants. Additionally, it allows us to consider uncertainties when the system model is not available, as the controller design is based on expert knowledge. The opposite case occurs in several control techniques that require at least linear system representations in controller design.

In this section, steps and design considerations of the fuzzy controller are presented.

### 3.1. Stage 1. Expert Analysis of the VSC Outer-Loop Control

In complex systems, typically the detailed mathematical model is harder to define. However, fuzzy mathematics allows the integration of expert knowledge in the model and, in that way, approximates the real system.

Consequently, this stage is oriented to analyse the expert knowledge about VSC functioning and how associate it with the control method's purpose. It contains three steps, which are described in the following:

#### 3.1.1. Step 1.1. Analysis of Micro-Grid Behaviour According to the Operation Mode

Micro-grids have two operating modes: connected and islanded from the power grid. A micro-grid has to operate in both modes to supply load adequately [36].

Energy storage devices are kept idle or charged during the grid-connected mode, depending on their charging status. Available distributed sources supply the loads, and the main power grid provides the deficit. In surpluses from distributed sources, this energy is stored or delivered to the main power grid.

During grid-islanded mode, the micro-grid contains distributed generators, storage devices, and loads, working separately from the main power grid.

In either micro-grid operating mode, storage devices must compensate for a deficiency in case of a power shortage.

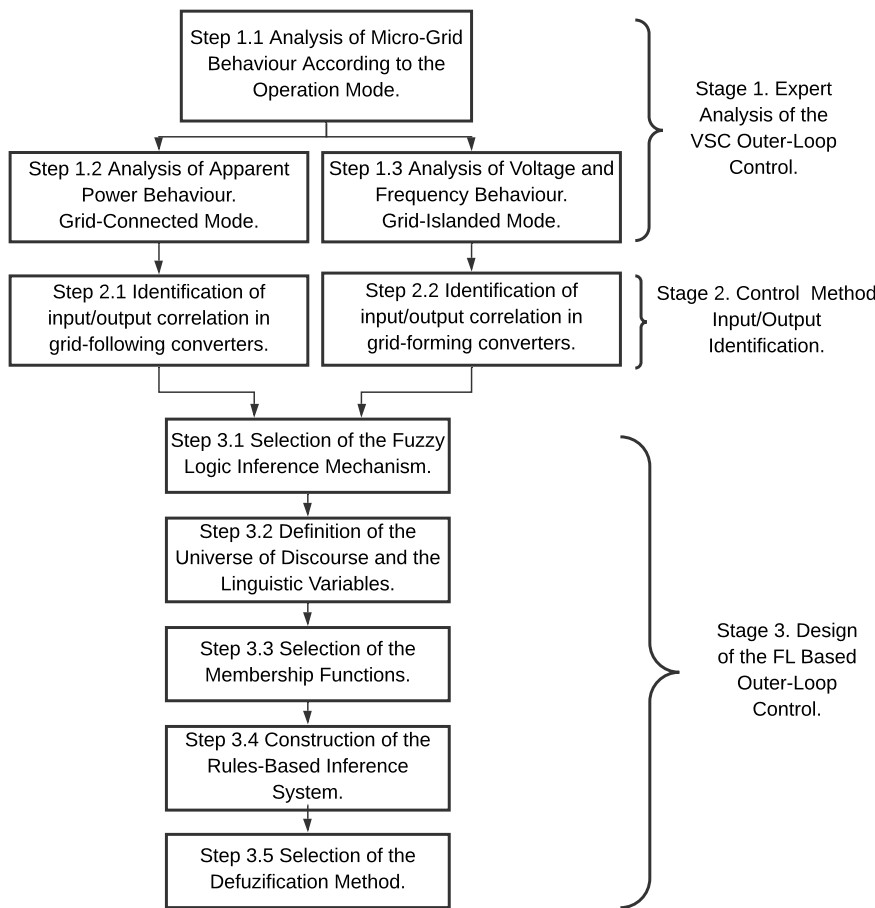

**Figure 3.** Proposed methodological approach.

### 3.1.2. Step 1.2. Analysis of Apparent Power Behaviour. Grid-Connected Mode

Control objectives of this operating mode are oriented to deliver the maximum power generated by renewable sources. In the case of storage devices, these are charged when the system has surpluses from local renewable sources. Consequently, the interests controlled are $P$ and $Q$.

### 3.1.3. Step 1.3. Analysis of Voltage and Frequency Behaviour. Grid-Islanded Mode

Control objectives of this operating mode are oriented to maintain adequate operating conditions. Consequently, the controlled variables are $v$ and $f$.

### 3.2. Stage 2. Control Method Input/Output Identification

The objective at this stage is to identify a correlation between the input and the output variables. In this case, the VSC is an interface element in charge of power management and grid efficiency improvement. Thus, grid operation depends on these converters' capacity to guarantee the assigned control objectives [9]. However, controller outputs are limited by the number of actuators that allow the necessary actions to set reference values for $P$, $Q$, $v$ and $f$.

The process of control input/output identification is composed of two steps:

### 3.2.1. Step 2.1. Identification of Input/Output Correlation in Grid-Following Converters

Grid-following converters adequately operate if there is a reference of voltage and frequency at the micro-grid. The proposed inputs for outer-loop based FLC in a grid-following converter are $P$ and $Q$. Quadrature and direct axis currents in the converter ($i_{d,q}$) are proposed as FLC outputs. In Figure 4, a graphical representation of the proposed outer-loop is presented.

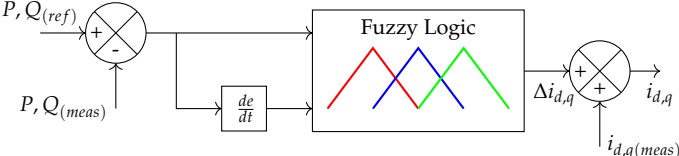

**Figure 4.** Outer-loop control based on fuzzy logic for grid-following converters.

### 3.2.2. Step 2.2. Identification of Input/Output Correlation in Grid-Forming Converters

Grid-forming converters operate in grid-islanded mode and work as an AC voltage source with a defined voltage and frequency. This system remains disconnected from the main grid; however, the converter sets micro-grid voltage and frequency references in case of a grid failure. In this case, references imposed by a grid-forming converter are used for the remaining converters, which operates as grid-following converters.

Based on previous reports, $v$ and $f$ are selected as inputs of the outer-loop based FLC. Additionally, as in the previous case, the converter currents in quadrature and direct axis ($i_{d,q}$) are proposed as FLC outputs. In Figure 5, a graphical representation of the proposed outer-loop is presented.

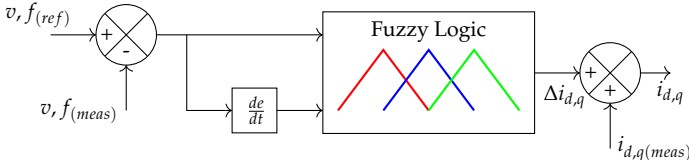

**Figure 5.** Outer-loop control based in fuzzy logic for grid-forming converters.

### 3.3. Stage 3. Design of the FL Based Outer-Loop Control

This stage is oriented to the design of the proposed FL-based outer-loop control. It is divided into five main steps, as shown in Figure 3.

The control objective is power management in grid-connected mode, considering direct and quadrature axis currents as controlled variables. The micro-grid voltage and frequency are controlled in grid-islanded mode, using direct and quadrature axis currents as controlled variables.

### 3.3.1. Step 3.1. Selection of the Fuzzy Logic Inference Mechanism

Fuzzy approaches are applied in complex nonlinear systems with unknown or poorly known mathematical models. There are two different types of fuzzy systems, Mamdani and Takagi–Sugeno–Kang (TSK); the premise of both systems' rules is the same, but the consequent in Mamdani needs a defuzzification step. This paper selects the Mamdani type by using linguistic variables to define the system's behaviour, considering the input and output relationships. Additionally, a precise system mathematical description is not required during the control design, and knowledge can be heuristically obtained [34,37].

### 3.3.2. Step 3.2. Definition of the Universe of Discourse and the Linguistic Variables

The universe of the discourse range is determined from the maximum and minimum physically allowed values of actuators and sensors and the nominal values of the input and output control method parameters. In grid-connected mode, the controller input value varies according to $P$ and $Q$ delivered from the renewable source to the network. In grid-isolated mode, it depends on the allowed operating ranges of $v$ and $f$.

Membership functions require the definition of linguistic variables that describe the probable values at each operating mode. The number of linguistic variables is related to the control method sensitivity and depends on the expert knowledge. In this case, as the input–output correlation is proportional, the same linguistic variables are chosen for inputs and outputs.

Ranges for linguistic variables are defined symmetrically from the zero-edge since no additional emphasis on any specific value is required. Finally, these functions have an intersection, which represents partial truth, to describe approximate reasoning. In this document, the interception point between membership functions is established as the intermediate point of the defined rank aimed to have a smooth intercept between the linguistic values and to avoid sudden changes, as in the whole truth problems.

### 3.3.3. Step 3.3. Selection of Membership Functions

Membership functions are chosen based on expert knowledge. However, the number of inputs and outputs required by the control method and the number of obtained rules can impair the computational effort.

The selection of membership functions varies depending on the system's behaviour; e.g., Gaussian and sigmoid functions are frequently used in processes with exponential changes. However, these two functions require a much longer processing time. Contrarily, linear functions such as trapezoidal and triangular types, significantly reduce the processing time and represent the proportion between the variation of input values and the corresponding degrees of certainty. Nevertheless, trapezoidal functions provide some degree of freedom for several values, corresponding to the degree of certainty in the membership function's upper base. The used membership functions are not limited to these basic functions, as they can be optimised considering the control method behaviour. Membership function structure can be modified from a better fit, initially considering the system's behaviour in the face of the control method actions obtained and the allowed error. However, this requires detailed knowledge of the system to translate this behaviour into the proposed functions.

### 3.3.4. Step 3.4. Construction of the Rule-Based Inference System

In grid-islanded mode, system behaviour is analysed to define the grid-forming converter FLC's knowledge base. In this case, it is observed that the system maintains voltage and frequency around the rated values, as defined in the control objective. For this reason, the inference system development is oriented to monitoring the error behaviour and the rate of change (derivative) of voltage and frequency. Analysis of the monitored variables establishes rules that define the behaviour.

System behaviour analysis is carried out in grid-connected mode to obtain a knowledge base used in the grid-following converter FLC. The power error behaviour is analysed by considering the variation in currents in the quadrature and direct axis. The relationship between powers and currents is linear and proportional. In this case, linguistic values with higher weights are chosen in case of small input changes; then, the system can be quickly brought to a new reference value during slight input variations. This characteristic is defined due to the frequent change perceived in renewable generation sources.

### 3.3.5. Step 3.5. Selection of the Defuzzification Method

Linguistic variables are converted back into numerical variables in the defuzzification stage, as the fuzzy controller output is based on the membership functions. After the rule evaluation for a specific input value, a composed and truncated area is obtained. This defuzzification method evaluates the acquired area and delivers the corresponding input.

In this paper, bisector is proposed as a defuzzification method. It requires low processing effort for area decomposition, avoids repeating areas, and deals with complex geometries.

## 4. Testing and Analysis of Results

### 4.1. Test System

The 400 V test system proposed in this paper consists of a variable load, two converter-integrated distributed generators and one storage device. The proposed micro-grid topology is presented in Figure 6.

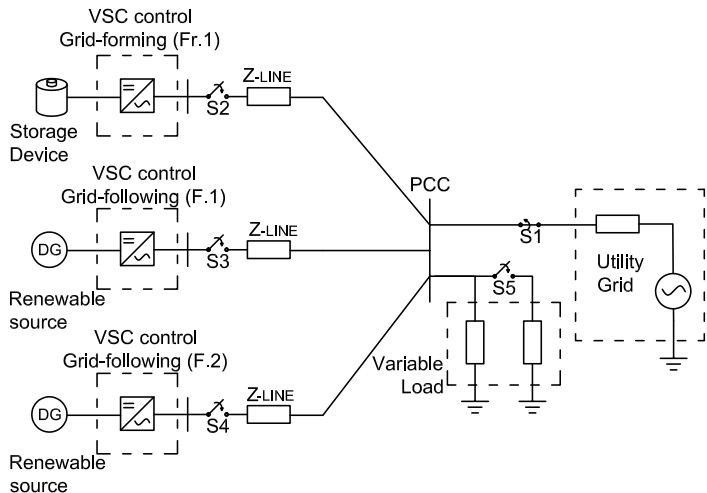

**Figure 6.** Test system.

Line parameters consider positive and zero sequence resistances of 0.423 and 0.3864 [$\Omega$/km]; positive and zero sequence inductances of $0.4265 \times 10^{-3}$ and $4.1264 \times 10^{-3}$ [H/km]; positive and zero sequence capacitances of $310 \times 10^{-9}$ and $7.751 \times 10^{-9}$ [F/km] and the line lengths is equal to 0.1 [km]. Fixed load is $300 + 145.4i$ [kVA] and variable load is $35 + 33.9i$ [kVA]. All DERs have the same rated power of 100 [kVA].

### 4.2. Testing Scenarios

The proposed scenarios are oriented to validate the VSC operation with fuzzy mathematics-based control methods in case of variations in load, three-phase faults, connection of other converters and change in micro-grid operating modes. This validates the VSC operation and the designed fuzzy controller. Tables 1 and 2 show the proposed testing scenarios.

**Table 1.** Analysed micro-grid scenarios.

| No. | Scenarios | Switching Times [s] | | | | | | |
|---|---|---|---|---|---|---|---|---|
| | | **0.2** | **1.5** | **2.0** | **2.2** | **4.0** | **5.0** | **5.2** |
| 1 | Connection of grid-forming converter | Fr.1 | | | | | | |
| 2 | Connection of grid-following converter | | F.1 | | | F.2 | | |
| 3 | Reference variation for active power | | | F.1 | | | F.2 | |
| | Reference variation for reactive power | | | | F.1 | | | F.2 |

**Table 2.** Connection times and disturbances.

| No. | Scenarios | Switching Times (s) | | | | |
|---|---|---|---|---|---|---|
| | | **0.8** | **2.5** | **6.0** | **7.0** | **8.0** |
| 4 | Three-phase fault (50 ms) | ✓ | ✓ | | ✓ | |
| | Three-phase fault (70 ms) | | | | | ✓ |
| 5 | Load change | | | ✓ | | |

### 4.3. Definition of the Fuzzy-Based Outer-Loop Control Method

The procedure for designing the fuzzy control method has been explained in Section 3. The characteristics selected for each controller component are described below.

#### 4.3.1. Universe of Discourse

The FLC outputs vary between [−1, 1] since the inner-loop requires current values in p.u. Similarly, FLC inputs are on a per-unit basis, and the universe of discourse for errors is defined as [−1, 1]. On the other hand, the derivative is not physically limited; for this reason, the universe of discourse is defined according to the maximum and minimum values of the variable. In this case, the derivative's maximum values range [−20,000, 20,000]. These take negative values, which allows the FLC to be used for storage systems charged from the network. These values are selected based on multiple simulations of the test system, which has allowed us to observe values taken by the variables under study.

#### 4.3.2. Membership Functions

This paper proposes triangular and trapezoidal membership functions after evaluating the relation between system inputs and outputs. Extreme values in universes of discourse are defined as trapezoidal functions representing huge inputs and outputs, and the other membership functions are triangular. In this way, the use of linear functions is maintained, reducing the computational effort, i.e., the processing time of this control stage is reduced. In Figures 7 and 8, the membership functions used for FLC are observed as proposed in Section 3.3.3.

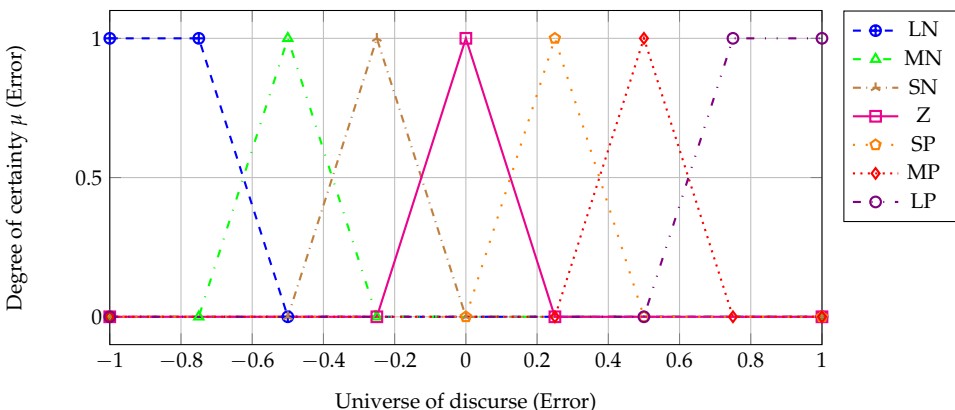

**Figure 7.** Membership functions for grid-following converters.

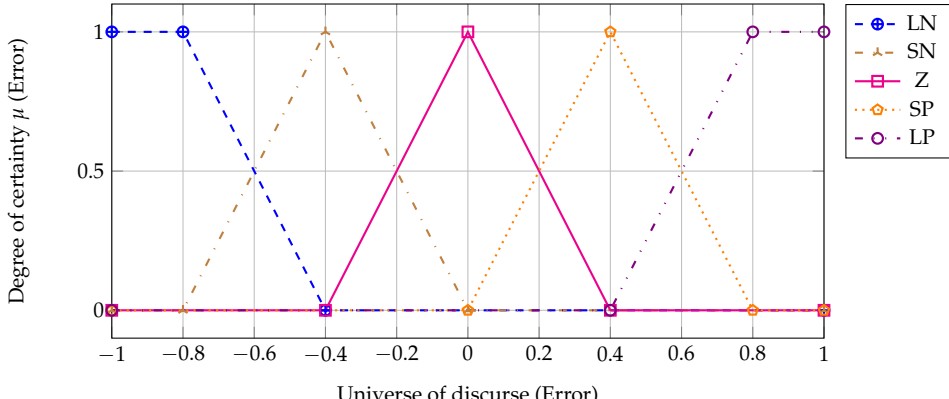

**Figure 8.** Membership functions for grid-forming converters.

#### 4.3.3. Definition of the Linguistic Values

The linguistic values selected for FLC in grid-islanded and grid-connected modes are defined as follows: the first is defined for variations around zero (*Zero* − *Z*), two for extreme

variations (*Large positive* $-$ *LP*, *Large negative* $-$ *LN*), and two more between zero and the extreme values previously defined (*Small positive* $-$ *SP*, *Small negative* $-$ *SN*). In grid-connected mode, two more functions are added (*Medium positive* $-$ *MP*, *Medium negative* $-$ *MN*). These linguistic variables are defined for inputs and outputs. The defined linguistic values are depicted in Figures 7 and 8. To reach these values, adjustments to eliminate oscillations and to improve response times were made.

#### 4.3.4. Definition of the Rule-Based Inference System

In grid-forming converters, the rules in Table 3 are obtained. The number of rules is defined by the combinations between the number of inputs and the membership functions chosen. In this case, twenty-five rules are defined.

**Table 3.** Rules for FLC of grid-forming converters. The rules are generically represented, since the relation $v_d - i_d$ and $v_q - i_q$ are similar.

| $\triangle i$ | | **Error Derivative $\dot{e}$** | | | | |
|---|---|---|---|---|---|---|
| | | **LN** | **SN** | **Z** | **SP** | **LP** |
| Error $e(t)$ | **LN** | Z | SN | LN | LN | LN |
| | **SN** | SP | Z | SN | LN | LN |
| | **Z** | LP | SP | Z | SN | LN |
| | **SP** | LP | LP | SP | Z | SN |
| | **LP** | LP | LP | LP | SP | Z |

For grid-following converters, the forty-nine rules in Table 4 are obtained.

**Table 4.** Rules for FLC of grid-following converters. The rules are generically represented, since the relation $P - i_d$ and $Q - i_q$ are similar.

| $\triangle i$ | | **Error Derivative $\dot{e}$** | | | | | | |
|---|---|---|---|---|---|---|---|---|
| | | **LN** | **MN** | **SN** | **Z** | **SP** | **MP** | **LP** |
| Error $e(t)$ | **LN** | LN | LN | LN | MN | SN | SN | Z |
| | **MN** | LN | LN | MN | SN | SN | Z | SP |
| | **SN** | LN | MN | SN | SN | Z | SP | SP |
| | **Z** | MN | SN | SN | Z | SP | SP | MP |
| | **SP** | SN | SN | Z | SP | SP | MP | LP |
| | **MP** | SN | Z | SP | SP | MP | LP | LP |
| | **LP** | Z | SP | SP | MP | LP | LP | LP |

### 4.4. Results and Analysis

Five scenarios have been considered to validate the proposed control algorithms. The following sections discuss the considerations for each scenario and the main findings.

#### 4.4.1. Scenario 1. Connection of a Grid-Forming Converter

In Figure 9, the system behaviour is observed in the case of the islanded operating mode considering both the FL- and PI-based control strategies. At 0.2 [*s*], the converter is connected to supply the load, following the voltage and frequency reference. It is observed that at 0.2 [*s*], both control strategies manage the system stabilisation when a grid-forming converter connection occurs. However, when the fuzzy logic-based control method is applied, the converter has higher oscillations, but the establishment time is similar to the

PI-based control method proposed in [24]. This condition can be improved by optimising membership functions.

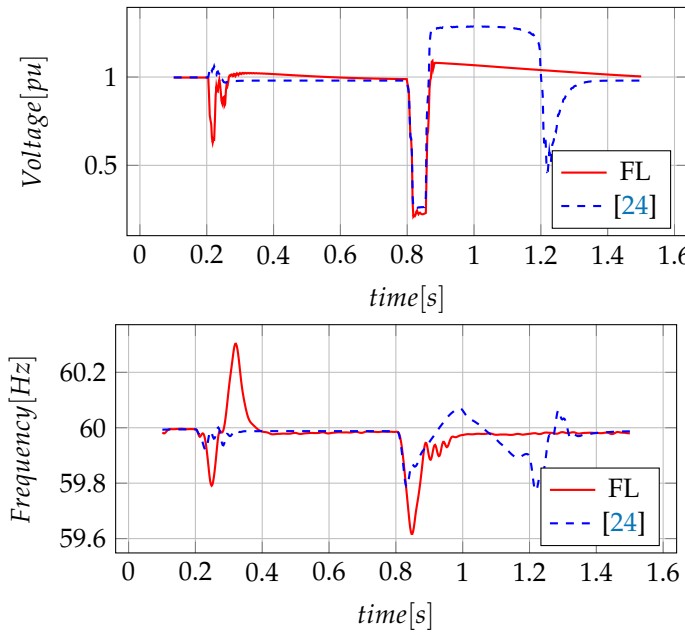

**Figure 9.** Measurement voltage and frequency in grid-forming converter.

4.4.2. Scenario 2. Connection of a Grid-Following Converter

In Figure 10, the disturbance in voltage and frequency signals is observed when a new converter is connected to the system. In this case, it is observed that both techniques reach their control objective, although oscillations in frequency are increased with fuzzy control application. However, the voltage signal has a drop more significant than 10% of the rating in case of application of the classical PI control technique proposed in [24]. In addition, the voltage establishment time is halved in the fuzzy-based control method application compared to the PI control strategy.

In Figure 11, a power disturbance at 1.5 [s] is observed when the grid-following converter is connected to the system. In this case, the control objective is accomplished with both strategies; however, in the case of the PI control method proposed in [24], it is observed that there are more significant variations in active and reactive power signals.

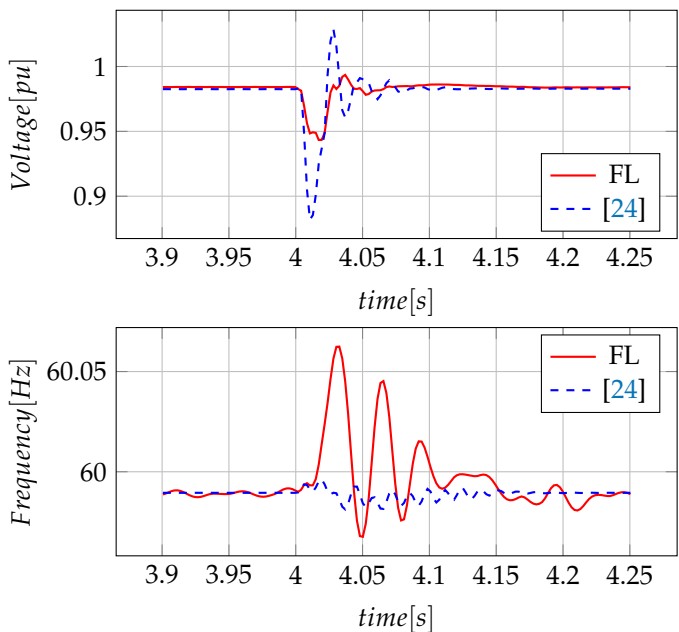

**Figure 10.** Measurement voltage and frequency of the system with connection of other converters.

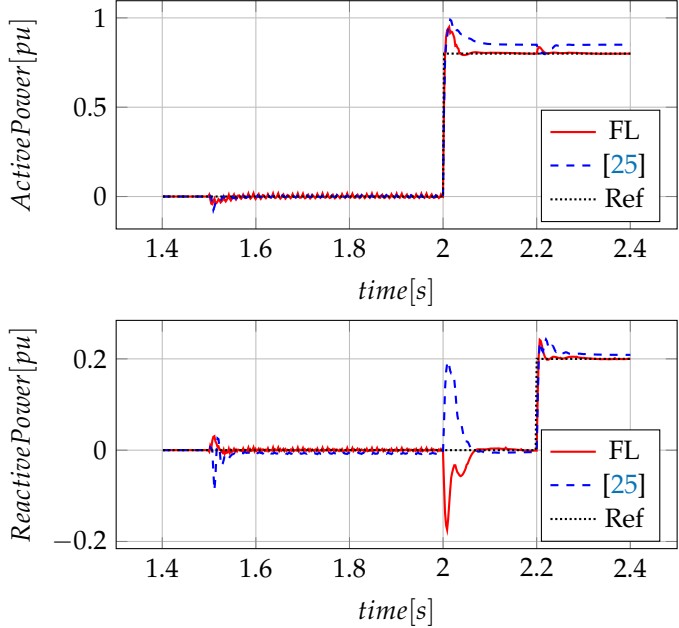

**Figure 11.** Converter connection and reference change of active and reactive power in grid-following converter.

### 4.4.3. Scenario 3. Power Reference Change in a Grid-Following Converter

In Figure 11, the tracking control of *P* and *Q* when the renewable source delivers power to the micro-grid in time of 2 [*s*] is presented. In this case, the micro-grid presents disturbances in reactive power signal when changes occur in the active power signal and vice versa; however, active power reference changes highly affect reactive power. Using the fuzzy-based outer-loop control method makes it possible to attenuate these peaks in case of changes in the generation reference, achieving a fast stabilisation and an adequate follow-up of the imposed reference. On the contrary, although the classical PI technique proposed in [25] manages to stabilise the system, in the case of not delivering power, the converter absorbs reactive power. Additionally, when it is required to provide power to the micro-grid, it delivers more than is required.

### 4.4.4. Scenario 4. Three-Phase Faults at the Micro-Grid

In this scenario, a three-phase fault at 0.8 [*s*] and a duration of 50 [*ms*] is considered, as it is observed in Figure 9. As a consequence of the three-phase fault, the micro-grid presents a high disturbance in the case of using a classical PI controller as proposed in [24]. When the three-phase fault is cleared, both techniques stabilise the system. However, when the PI control technique is applied, sustained voltage swell is observed for an approximate time of 0.3 [*s*], until it is restored to the rated value. In this case, the fuzzy logic technique manages high voltages and generates a gradual reduction until the signal reaches the rated voltage value.

On the other hand, when a three-phase faults occurs, frequency is severely affected. In the case of the fuzzy logic-based outer-loop control method, more significant initial variations are observed in comparison to the classical control method proposed in [24]. However, the establishment time is reduced by half compared to the classical control technique. Moreover, when the three-phase fault is cleared, frequency oscillations are reduced using fuzzy logic-based control compared to the classical control technique.

### 4.4.5. Scenario 5. Load Change

In the Figure 12, a change in the power reference of a grid-following converter is observed at 6 [*s*]. In this case, the load change in grid-following converters generates oscillations in power signals, as seen in Figure 12, because these converters try to deliver the required power. A grid-following converter is configured to deliver the power adjusted at the reference, considering its physical and operational parameters; therefore, in case of load increases, an energy management system is required, which changes the reference of these converters. However, it is observed that the power tracking control objective is achieved. Additionally, both techniques manage to follow the power reference imposed by generation source, where oscillations in the signals are attenuated by the control technique based on fuzzy logic, as shown in Figure 12.

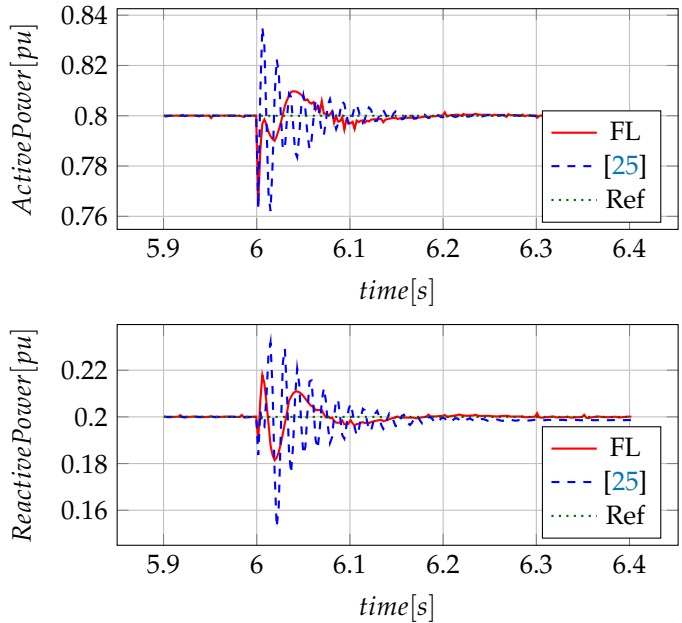

**Figure 12.** System disturbances seen from grid-following converter 2 (load change).

In the analysed PI control strategies, the gain tuning is an issue since conventional trial-error processes can produce sub-optimal solutions that affect the system's dynamical performance.

The integral of time-weighted absolute error (ITAE) and the integral of absolute error (IAE) are used as the performance index, and in this way, the fuzzy controller's effectiveness is demonstrated. Results are shown in Tables 5 and 6.

**Table 5.** IAE and ITAE for grid-forming. Voltage ($v$) and frequency ($f$) control.

| Variable | | Control Strategy | |
|---|---|---|---|
| | | Fuzzy | PI |
| IAE | $v$ | 0.926 | 0.853 |
| | $f$ | 0.197 | 0.311 |
| ITAE | $v$ | 6.093 | 4.685 |
| | $f$ | 1.014 | 1.904 |

**Table 6.** IAE and ITAE for grid-following 1. Active Power ($P$) and Reactive Power ($Q$) control.

| Variable | | Control Strategy | |
|---|---|---|---|
| | | Fuzzy | PI |
| IAE | $P$ | 0.459 | 0.795 |
| | $Q$ | 0.388 | 0.548 |
| ITAE | $P$ | 3.566 | 5.412 |
| | $Q$ | 3.071 | 3.726 |

Finally, the sampling time used in the simulations is $5 \times 10^{-5}$ s.

## 5. Conclusions

A fuzzy mathematics-based outer-loop control method is proposed and validated to avoid using the detailed mathematical model of the micro-grid. As observed in this paper, general knowledge related to the micro-grid behaviour is required. The fuzzy-based control method has advantages compared to the classic PI control in dealing with disturbances. Additionally, a low settling time is achieved by following the detailed guides proposed in this document. It is demonstrated with a straightforward converter controller design, used in grid-connected and grid-islanded operating modes, that the control objectives are achieved in both cases. In addition, to test the control method reliability, different fault scenarios are proposed. The proposed controller succeeds in overcoming faults and restoring the system in a shorter time than the classic control strategies that require more significant adjustment efforts for the same system. This type of controller can be easily implemented in nonlinear systems as voltage source converters.

The approach proposed in this paper uses Mamdani-type inference on a fuzzy-logic-based control method applied to converter-connected distributed generation and storage devices in micro-grids. The primary method limitations are associated with selecting the values of the universe of discourse and the type of membership functions. The optimisation of these parameters allows us to obtain a better controller performance. Other fuzzy inference drawbacks are observed in the intersected boundaries and an improper transition region between the inference boundaries of non-intersected.

In future work, robustness can be evaluated by varying the converter parameters to trigger control errors. This allows us to differentiate converters according to the number and type of errors uncovered and provides information to solve the identified problems. Parameter variation helps us to consider uncertainties in the system modelling. Furthermore, optimal control can be proposed as an objective function defined by minimising the difference between the system response and the reference signal. Variables have to be restricted by values with physical meaning according to the system size. Finally, the design

of a Takagi–Sugeno-type controller has to be studied to ensure the system stability and to consider the available micro-grid mathematical model.

**Author Contributions:** Conceptualization, M.B.-L. and J.M.-F.; methodology, M.B.-L. and J.M.-F.; software, L.C.; validation, L.C. and M.B.-L.; formal analysis, L.C., M.B.-L. and J.M.F.; investigation, L.C.; writing—original draft preparation, L.C.; writing—review and editing, M.B.-L. and J.M.-F.; supervision, M.B.-L.; project administration, J.M.-F.; funding acquisition, M.B.-L. and J.M.-F. All authors have read and agreed to the published version of the manuscript.

**Funding:** This paper is a result of projects 111077657914—contract 031-2018 and 111085271060—contract 774-2020, funded by the Colombian Ministry of Science, Technology, and Innovation (Minciencias) and developed by the ICE3 Research Group at the Universidad Tecnológica de Pereira (UTP).

**Data Availability Statement:** Data is available under formal request to Lorena Castro and Maximiliano Bueno-López.

**Conflicts of Interest:** The authors declare no conflict of interest. The funders had no role in the design of the study; in the collection, analyses, or interpretation of data; in the writing of the manuscript, or in the decision to publish the results.

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
