# Peer review of "Fuzzy Mathematics-Based Outer-Loop Control Method for Converter-Connected Distributed Generation and Storage Devices in Micro-Grids"

_computation, doi:10.3390/computation9120134_

Round 1

Reviewer 1 Report

The authors proposed a fuzzy-based outer-loop control and validate it to avoid using the micro-grid’s detailed mathematical model and the converter itself. I cannot not find a significant trace of why the authors employed fuzzy concept while we cannot find the superiority in comparison with the deterministic cases at least from the provided results? It is important that the author demonstrate the effectiveness of using fuzzy concept based on provided data while the data base is poor and not enough. Also, a fair comparison with other established works has been missed.

Reviewer 2 Report

The current paper proposes a multivalued logic-based strategy to avoid using simplified micro-grid models. Simulation results validate the proposed control strategy’s proper functioning, demonstrating its capabilities for real field applications

Comments to author:

- The authors can add the steps of implementing the algorithms. The theoretical part can be better detailed. The steps will be in the benefit of the readers, maybe they’ll help the readers to implement the proposed algorithm.

- Please add more details of how the theory from the previous sections is applied is applied in the results section.

- Please specify the sampling time.

- Please specify the controller parameters and how the parameters were obtained.

- The authors could add a paragraph with the advantages and the disadvantages of the proposed method.

- The state of the art should be improved with more references, maybe the author could add the following publications:

o Hybrid Data-Driven Fuzzy Active Disturbance Rejection Control for Tower Crane Systems, European Journal of Control, vol. 58, pp. 373-387, 2021.

o Multi-Agent-Based Data-Driven Distributed Adaptive Cooperative Control in Urban Traffic Signal Timing, Energies, vol. 12, no. 7, pp. 1–19, 2019.

- Please add more details and comment the obtained results from chapter 4.4.

Reviewer 3 Report

Dear Authors,

Based on the 1st round review of the manuscript entitled Fuzzy mathematics-based outer-loop control for converter-connected distributed generation and storage devices in micro-grids, the reviewer has the following comments:

  1. How you can validate the reliability in the proposed method? Please explain in more details about it in the revised manuscript.
  2. Robustness is an important factor to control of industrial applications. Which technique can be used to validate the robustness? Please highlighted in the revised manuscript.
  3. How you can optimize the membership functions and the other fuzzy parameters? Please highlighted in the revised manuscript.
  4. How you can testing the power of your technique against to uncertainties? (explain in the revised manuscript.)
  5. Please explain about the future work in the conclusion.
  6. What is the proposed method limitations? please explain in the conclusion and highlighted in the revised manuscript.

Regards,

Round 2

Reviewer 1 Report

It is accepted in the present format.